# Focus on the Small GTPase Rab1: A Key Player in the Pathogenesis of Parkinson’s Disease

**DOI:** 10.3390/ijms222112087

**Published:** 2021-11-08

**Authors:** José Ángel Martínez-Menárguez, Emma Martínez-Alonso, Mireia Cara-Esteban, Mónica Tomás

**Affiliations:** 1Department of Cell Biology and Histology, Biomedical Research Institute of Murcia (IMIB-Arrixaca-UMU), Medical School, University of Murcia, Campus Mare Nostrum (CMN), 30100 Murcia, Spain; emma@um.es; 2Department of Human Anatomy and Embryology, Medical School, Universitat de Valencia, 46010 Valencia, Spain; micaes2@alumni.uv.es (M.C.-E.); monica.tomas@uv.es (M.T.)

**Keywords:** Rab1, GTPases, Parkinson’s disease, secretory pathway, Golgi fragmentation, autophagy, α-synuclein, LRRK2

## Abstract

Parkinson’s disease (PD) is the second most frequent neurodegenerative disease. It is characterized by the loss of dopaminergic neurons in the substantia nigra and the formation of large aggregates in the survival neurons called Lewy bodies, which mainly contain α-synuclein (α-syn). The cause of cell death is not known but could be due to mitochondrial dysfunction, protein homeostasis failure, and alterations in the secretory/endolysosomal/autophagic pathways. Survival nigral neurons overexpress the small GTPase Rab1. This protein is considered a housekeeping Rab that is necessary to support the secretory pathway, the maintenance of the Golgi complex structure, and the regulation of macroautophagy from yeast to humans. It is also involved in signaling, carcinogenesis, and infection for some pathogens. It has been shown that it is directly linked to the pathogenesis of PD and other neurodegenerative diseases. It has a protective effect against α–σψν toxicity and has recently been shown to be a substrate of LRRK2, which is the most common cause of familial PD and the risk of sporadic disease. In this review, we analyze the key aspects of Rab1 function in dopamine neurons and its implications in PD neurodegeneration/restauration. The results of the current and former research support the notion that this GTPase is a good candidate for therapeutic strategies.

## 1. Introduction to PD Pathogenesis

PD is the second most frequent neurodegenerative disease, affecting around 4 million people around the world, with a prevalence and incidence rate of 108–257/ and 11–19/100,000 per year, respectively [1]. Even though a detailed analysis of the disease and almost all of its variations has been provided before, only palliative treatments for it exist. Unfortunately, diagnosis is almost exclusively based on symptomatology, while the only accurate diagnosis can be made by the post-mortem analysis of the brain [2,3]. Apart from classic motor symptoms such as rigidity or akinesia, non-motor symptoms have also been described, including dementia and other specific cognitive disorders, as well as gastrointestinal problems [4,5,6].

This disease is characterized by the loss of dopaminergic neurons in the substantia nigra pars compacta (SNpc), which affects many cerebral circuits controlling motor movements and other functions [7]. The cytopathological hallmark of the survival nigral neurons is the formation of insoluble deposits, called Lewy bodies (LBs), as well as the less studied Lewy neurites and pale bodies [8]. A recent electron microscopic analysis showed that these bodies contain filamentous materials, lipids, and altered organelles, such as mitochondria, lysosomes, and autophagosomes [9]. Although more than one hundred proteins have been identified, they are primarily composed of insoluble forms of α-synuclein (α-syn) [10]. This small protein (14 kDa) is detected in many compartments, such as cytosol, the nucleus, and mitochondria, although it is functionally associated with synaptic vesicles in presynaptic terminals. It is encoded by the *SNCA* gene and belongs to a well-conserved family of synaptic proteins formed by β-, γ-, and α-syn [11]; however, only the *α* isoform is able to form toxic fibrillary structures. Its exact function is unclear, although it has been suggested that it regulates synaptic vesicles’ exocytosis and recycling and the formation of fusion pores [12]. Cytosolic *α*-syn is able to sense membrane curvature and bind negatively charged phospholipids present in the membranes of synaptic vesicles [13]. It has been proposed that it acts as a chaperone for the assembly of SNARE complexes (complexes mediating the specific fusion of compartments) at the terminal end, regulating the release of dopamine and avoiding non-specific interactions [14]. This protein is a major player in PD pathogenesis [10]. Monomers of α-syn can assemble to form beta-sheet-rich oligomers, protofibrils, and amyloid fibrils, which can aggregate to form LBs. These can alter many cellular processes, such as secretory and endocytic trafficking, autophagy, mitochondrial and nuclear functions, and proteasome degradation, which may induce cell stress and death [15,16]. The internalization of aggregates of misfolded α-syn might contribute to the spread of the pathology between neurons in a mechanism resembling prions [17]. Age is a key factor in *α*-syn aggregation and propagation [18].

The reason that dopaminergic neurons are particularly affected remains unclear; multiple alterations related to mutations in key proteins involved in protein homeostasis, mitochondria function, lysosome/autophagy pathways, dopamine synthesis, metabolism, the immune system, and inflammation have been proposed [19,20]. Many experimental data strongly support the notion that defective intracellular transport could be a major cause of nigral neuron death [21,22,23]. The axon of each nigral dopamine neuron travels a very long way to the striatum (the mesostriatal pathway); there, it branches extensively and establishes thousands or even millions of synapses with medium spiny neurons [24]. Thus, these neurons need to maintain “well-oiled” transport machinery to maintain a high flux of secretory proteins (and concomitantly a recycling apparatus), and its failure may trigger cell death.

The cause of sporadic PD is unknown, but 5–15% of cases are familial. Multiple mutations present in family members with PD are known, including autosomal dominant (*SNCA*, *LRRK2*, *VPS35*) and recessive (*PARK2*, *PINK1*, *DJ-1*) forms [25], although many additional risky loci have been identified [26,27]. The overexpression (duplication or triplication) or mutation of the *SNCA* gene is known to be a cause of familial PD [25,28,29]. Most *SNCA* point mutations are associated with the early onset of disease. These mutations and the greater production of α-syn result in aggregation, leading to the formation of LBs. The *SNCA* locus is also a risk for sporadic PD.

However, the most common causes of familial PD are mutations in leucine-rich repeat kinase 2 (*LRRK2*), and this is a strong risk factor for sporadic disease [30]. This is a large multi-domain protein (280 kDa) with a GTPase domain, serine/threonine kinase domains, and other domains involved in protein–protein interactions [31]. This kinase interacts with many molecules, including Rab GTPases. Mutations in these domains may lead to a hyperactive protein with toxic effects. LRRK2 plays multiple roles in cells, including in vesicular trafficking, endosomal transport, autophagy, cytoskeleton dynamics, ciliogenesis, neurite growth, mitochondrial morphology, and mitochondrial calcium regulation [31,32,33,34,35]. In nigral neurons, it is specifically involved in vesicle formation and docking, as well as ER-to-Golgi and Golgi-to-membrane transport and the internalization of dopamine receptors at the synapse [36,37].

VPS35 is also involved in transport, specifically in endosome–TGN retrograde transport mediated by the retromer complex. A PD-related mutation of this retromer subunit enhances the LRRK2-dependent phosphorylation of Rabs [38]. Mutations in the *PARK2* (parkin), *PINK1*, and *DJ-1* genes affect mitochondrial functions/homeostasis [39].

Many intracellular processes are altered in dopaminergic neurons, which can lead to their death [40]. All/most of these processes are connected and several molecules can be key regulators and monitors all of these routes. Thus, knowledge of these molecules is very important in order to understand neurodegeneration and could be the target of future therapeutic strategies. Rab GTPases are important regulators of membrane traffic. In the present review, we analyzed one of these small GTPases, Rab1. This protein has long been known as a regulator of the early steps of secretory traffic. It was a surprise when it was found to be related to *α*-syn toxicity. Furthermore, it was demonstrated to play a key role in autophagy and many other processes, such as unfolding protein response (UPR), signaling and carcinogenesis [41], glucose homeostasis [42], and infection with some parasites (such as Legionella pneumophila) [43,44].

Surviving nigral neurons have been found to overexpress this GTPase in a cellular model of PD and human samples [45,46]. Now, it is evident that it is a key molecule in PD and other neurodegenerative diseases. Thus, the alteration of Rab1-dependent ER–Golgi transport is a pathogenic mechanism involved in common types of familiar amyotrophic lateral sclerosis [47]. One of these disease-related mutants (FUS) was found to impair autophagy and, importantly, Rab1 overexpression plays a protective role [48]. Rab1 is also involved in the transport and processing of amyloid precursor protein (APP) in Alzheimer’s disease [49]. Here, we analyze the molecular/functional aspects of the molecular biology of Rab1 directly related to PD.

## 2. Rab GTPases

Rab proteins are the largest subfamily of GTPases, with more than 60 members of approximately 25 kDa, sharing around 30% sequence identity. Their functions include the maintenance of the shape and dynamics of intracellular membranes and the establishment of membrane identity [50]. Rabs are involved in the formation, cytoskeleton-dependent transport, docking, and fusion of transport vesicles. To fulfill their functions, Rabs interact with many molecules, called effectors, which include coat proteins, tethering factors, cytoskeleton elements, etc. They were first discovered in nervous tissue, where vesicular transport is essential for dendrite and axon organization as well as synaptic transmission [51]. Each Rab or group of Rabs localizes in a particular membrane and regulates a specific transport step, including bidirectional ER–Golgi trafficking, intra-Golgi transport, endolysosomal routes, and autophagy [52,53]. In addition, they also regulate signal transduction, cell survival, and development [54].

Rabs are considered molecular switches and cycle between an active GTP-bound form and an inactive GDP-bound form [55] (Figure 1). In their active form, they are able to interact with many effector proteins, regulating multiple cellular processes. Their activity is regulated by guanine nucleotide exchange factors (GEFs), which promote the exchange of GDP to GTP and GTPase-activating proteins (GAPs); these activate the intrinsic low GTPase capability of Rabs. Inactive GDP-bound Rabs are found soluble in the cytosol, whereas, in their GTP-bound active forms, they are located in cellular membrane joints by geranylgeranyl groups. This binding is regulated by GDP dissociation inhibitor (GDI), which removes inactive GDP-bound Rabs from membranes. It has been proposed that GDI displacement factors (GDF) at the membrane inhibit this complex and maintain the association of Rabs with membranes. The association with membranes is also sensible to lipid composition, as it senses the curvature elastic energy stored in the membrane [56]. Rab activity is also finetuned by phosphorylation. This post-translational modification can affect the mode of interaction between Rabs and their partners [57]. Interestingly, a serine/threonine kinase that is specific for some Rabs is the above-mentioned LRRK2.

Rabs participate in many aspects of neuron life, including development, the polarized growth of neurites, endocytosis at the neuron end, axonal transport, the exocytosis of synaptic vesicles, and the dynamics of neurotransmitter receptors [58,59]. A few Rabs are selectively enriched in neurons, i.e., Rab3A (synaptic vesicle exocytosis), Rab8 (dendrite-specific transport), and Rab23 (neuron development and endosomal transport), all of which are associated with the post-Golgi transport step [60]. Most neurodegenerative diseases, including PD, are related to vesicular transport alterations mediated by Rabs [53,61,62]. Several Rabs (Rab8B, Rab11A, Rab13, and Rab39B) regulate the toxic aggregation of α-syn [63]. The LRRK2-dependent phosphorylation of Rab35 has been found to be involved in *α*-syn propagation between neurons [64]. The overexpression of Rab7 reduces *α*-syn aggregates by enhancing autophagy [65]. Wild-type *α*-syn physically interacts with Rab3A/B/C, Rab4B, Rab6A, Rab8A, Rab15, and Rab35 [66]. The PD-related mutant A30P *α*-syn interacts with Rab3A, Rab5, and Rab8, therefore affecting synaptic vesicle trafficking, endocytosis, and α-syn transport [67]. In a few cases, Rabs are the direct cause of the disease [51]. This is the case for Rab39B, which is involved in α-syn homeostasis, endosomal traffic, neurite outgrowth, synaptic maturation, and autophagy [50]. Rab39B mutation causes a rare form of early-onset PD with an α-syn pathology [68].

## 3. Rab1

Rab1 is one of the five Rabs present in all eukaryotes and is considered a housekeeping Rab. In yeast, Rab1 is encoded by the YTP1 gene. Two isoforms have been described, Rab1A and 1B, which share 75% and 66% amino acid identity with Ytp1, respectively, with 92% identity between them [69,70]. Both isoforms are expressed in all tissues, but there are important differences in their levels of expression [71]. The functional dissimilarities between these isoforms have not been exhaustively analyzed but there are differences in the interactions with coated vesicles and subcellular distribution (see below) and the role in the secretory flux of some proteins [72].

Rab1 interacts with many effectors related to membrane traffic, including p115 [73], GM130 [74,75], golgin 84 [76], and giantin [77]. Another effector of unknown function is Iporin, a ubiquitous protein that is highly expressed in the brain and that also interacts with GM130 [78]. Additionally, it interacts with members of the MICAL family and could form a link between membrane traffic and the cytoskeleton [79]. Rab1A and B interact with Golgi phosphoprotein 3 (GOLPH3), a protein involved in post-Golgi transport that is considered a Golgi oncoprotein and associated with several types of cancer [80].

As with all Rabs, GEFs, GAPs, GDI, and GDF regulate the association of Rab1 with membranes [55]. Two well-conserved GEFs for Rab proteins have been identified; these are named Transport Protein Particle (TRAPP) III and II (TRAPPI is now considered to be the core subcomplex of the other two). These regulate the early and post-Golgi transport steps, respectively. The multicomplex TRAPIII specifically activates Rab1, although TRAPPII has some activity on this Rab of unknown physiological significance [81]. In yeast, the membrane anchoring of TRAPIII is mediated by the Trs85 subunit, which is also necessary for Ypt1 activation [82]. The Bet5, Trs31, and Bet3 subunits are also important for this activity [83]. In mammals, TRAPPC4 (the ortholog of yeast Trs23) is a core component of the TRAPP complexes and is also essential for the GEF activity of Rab1 [55]. TRAPPC8 (which shares some characteristics with yeast Trs85) is also important for Rab1 activation [81]. TRAPPIII is located in COPII vesicles and Golgi. Its location in COPII vesicles depends in the interaction of the TRAPP core and the COPII component Sec23 [84,85]. TRAPPIII is also located in Atg9 vesicles and pre-autophagosomal structures in a process involving the Trs85 subunit [86] (see below for its role in autophagy). Interestingly, variants of TRAPPC4 are associated with human neurological disorders [87].

GAPs for Ypts are named GYPs in yeast and TBC1 domain proteins in other species due to all of them sharing this domain [88]. The yeast Gyp8 is a GAP for Ypt1, which is present in the ER. Gyp1, another GAP for Ypt1, is located in the early Golgi [89]. Their location in membranes depends on Arf1 GTPase, which recruits Gyp1 inactivating Ypt1 [90]. This sequential gradient of activation/inactivation of GTPases is important for maintaining compartment identity, ensuring cisternae maturation, and guiding the secretory traffic [91,92]. Gyp1 also plays a role in autophagy. It regulates the disassembly of Ytp1–Atg1 complexes, as well as the interaction of cargo receptor and Atg8 proteins in selective autophagy [93]. In humans, 38 Rab GAPs have been identified, with TBC1D20 (Gyp8 in yeast) being the specific (but not exclusive) GAP for Rab1. It regulates the exit from the ER of the secretory cargo, as well as the Golgi architecture [94]. It is also involved in the maturation of autophagosomes [95]. Interestingly, this study showed that TBC1D20-deficient mice display an altered autophagy flux in neurons, resulting in adult-onset motor dysfunction.

Less is known about the role of GDF and GDI in the location of Rab1. Prenylated Rab acceptor 1 (PRA1) (the mammalian homolog of yeast Yip3) has GDF activity for Rab1 (and other Rabs) [96] and is involved in its association with membranes. Interestingly, PRA1 interacts directly with α-syn [97]. Several well-conserved isoforms of GDI have been described that bind a wide spectrum of Rab proteins, with *α*-GDI being restricted to the brain [98]. GDI is involved in the retrieval of Rab1 from Golgi membranes [55].

## 4. The Role of Rab1 in the Secretory Pathway

It is well known that newly synthetized proteins and lipids move from the ER to the Golgi in COPII-coated vesicles, while retrograde transport is mediated by COPI-coated vesicles. The role of COPI-coated vesicles in the anterograde (i.e., ERGIC to Golgi and intra-Golgi transport) is under discussion. In contrast with yeast, in mammals, there is an intermediate compartment between the two organelles called ERGIC. The origin and role of these membranes are not completely understood. It is likely that it is formed by the fusion of ER-derived vesicles, which then move in block to the Golgi. Rab1 locates to the early membranes of the secretory pathway—more specifically, ERGIC and the cis-Golgi cisternae—but not in the ER [99]. In vivo experiments have shown that Rab1A is located in dynamic tubules with bidirectional movements, whereas Rab1B is concentrated in vesicular stationary elements [100], although these differences might be cell-type-specific. Rab1A-positive tubules exclude COPI and p58 (an ERGIC marker) [101], whereas Rab1B elements allow COPI budding [102].

Early studies support the notion that Rab1/Ypt1 is involved in the bidirectional traffic between ER and Golgi traffic (Figure 2A and Figure 3), and its location is highly dynamic in living cells [55]. It has been implicated in regulating COPII vesicle formation and subsequent tethering and fusion with Golgi/ERGIC membranes [83,103,104,105]. Thus, GDP-bound Rab1A and Rab1B (inactive forms) inhibit ER to Golgi transport [106]. Rab1B interacts with the COPII components Sec23, Sec24, and Sec31 [107]. Recent studies support the notion that cargo leaves the ER in Rab1-dependent carriers, whereas COPII components remain in stable ERES even when the cargo is released [108]. The Rab1 present in COPII vesicles may recruit p115, tethering these vesicles to the ERGIC membranes; similarly, Ypt1p interacts with Uso1, the yeast homolog of p115 [103]. The Rab1–p115 complex may anchor these elements to the cis-Golgi by interacting with GM130-GRASP65 [74]. The complex Rab1–p115 might be also involved in the tethering of COPI vesicles via giantin [77]. GM130, giantin, and p115 belong to the golgin family, which are Golgi matrix proteins characterized by extensive coiled-coil domains that, together with Rabs, organize Golgi membranes and transport [109]. Rab1 also interacts with another golgin, golgin-84, which is involved in the lateral linking of Golgi stacks, forming a continuous ribbon [110]. This interaction is necessary to maintain the Golgi structure and protein transport [111].

Ypt1 is also involved in COPI-dependent retrograde transport from the Golgi [112]. Ypt1 interacts with COPI [113]. In contrast with Rab1A, Rab1B regulates COPI coat recruitment to ERGIC and Golgi membranes because it is necessary for the recruitment of Golgi-specific brefeldin A-resistance guanine nucleotide exchange factor 1 (GBF1), which is the GEF that is specific for Arf1 [102], a GTPase involved in the recruitment of COPI components to membranes. In addition, Ypt1/Rab1 interacts with different subunits of the conserved oligomeric Golgi (COG) complex [114], which is involved in the tethering of vesicles during intra-Golgi retrograde transport.

Rab1 has recently been detected in the membrane of immature secretory granules, where it is recruited by Rab11 and might be involved in their maturation [115]. Whether this a general mechanism or restricted to the salivary glands of Drosophila remains to be clarified.

Rab1 also regulates the association of transport carriers and the cytoskeleton. Rab1 at the ERGIC can recruit WASP Homolog Associated with Actin, Golgi Membranes and Microtubules (WHAMM), which stimulates Arp2/3-mediated actin polymerization and interacts with microtubules, inducing membrane deformation [116]. The interaction of WHAMM with these cytoskeleton elements mediates membrane tubule elongation and formation, respectively [117]. Rab1 is also involved in the recruitment of the motor protein kinesin KifC1 to endocytic vesicles [118]. In addition, an effector of Rab1, GM130, is involved in microtubule polarization at the cis-Golgi [119]. Thus, Rab1 is a regulator of Golgi morphology and positioning.

Rab1 may also have specific roles in neuron trafficking. Most dendrites (80%) lack a Golgi outpost but contain ERGIC elements in close contact with early endosomes, forming a so-called satellite [120]. It has been proposed that, at this level, proteins and lipids are delivered from the ER to the synaptic spine membrane, using these two compartments by bypassing the Golgi [121]. Rab1-positive ERGIC tubules move to the growth cones of neurite-like processes in differentiated PC12 cells, a common model used for studying neuronal differentiation [101]. Thus, Rab1 may have a specific role in neuron development. In vivo experiments have shown that Rab1 tubules move bidirectionally between the central body and neurites connecting central Golgi and peripheric ERES. Rab1 is also involved in the formation of cell protrusions (i.e., neurites). It forms part of a conserved signaling pathway where the KDEL receptor stimulates the Golgi-resident monomeric G*α*o, which uses Rab1 (and 3) for the promotion of secretion to form protrusions [122].

There is a link between Rab1-dependent traffic and ER homeostasis. Protein aggregation induced by the pesticide rotenone decreases the levels of Rab1 in hippocampal neurons, which may impair secretory traffic and, as a consequence, trigger an unfolded protein response (UPR) [123]. In yeast, Ypt1 plays an important role in the regulation of UPR by stabilizing unprocessed HAC1 RNA under normal growth conditions, avoiding UPR activation when there are no stress conditions [124]. In fact, Ypt1 can switch molecularly and functionally from GTPase to a molecular chaperone under heat stress conditions, enhancing the resistance of the cell under these conditions [125].

## 5. Rab1 as a Regulator of Autophagy

Macro-autophagy is an important mechanism for the maintenance of cellular homeostasis through the degradation of damaged organelles or protein aggregates, the recycling of materials in the absence of nutrients or stress conditions, and even the elimination of pathogens. Apart from macroautophagy, the lysosome is also involved in two other mechanisms of autophagy: chaperone-mediated autophagy and microautophagy. In the first type, proteins bearing the KFERQ sequence motif are recognized by heat shock cognate 71 kDa protein (HSC70) and target lysosomal-associated membrane protein 2 (LAMP2) [126,127]. In the second type, a small portion of the cytoplasm is engulfed by invaginations of the lysosomal membranes, leading to the formation of small internal vesicles [128].

Macro-autophagy (hereafter autophagy when not specified) involves the formation of double-membrane autophagosomes sequestering cytoplasmic portions that will fuse with lysosomes. Although autophagy protects cells against stress, an excess may induce cell death. It has been considered a bulk mechanism, but it can be also a highly selective process where specific receptors recognize specific cargo [129]. This is the case for the degradation of fibrillar aggregates (aggrephagy) or mitochondria (mitophagy), among others. In non-dividing cells such as neurons, it is an important mechanism for maintaining cellular homeostasis. However, few autophagosomes are found in normal neurons, and special mechanisms, such as engulfing Golgi cisternae, may cooperate in degrading materials [130]. The consequence of the failure of autophagy is the accumulation of damaged organelles and proteins, which may lead to neurodegeneration [131].

The initial elements of autophagy are small structures called phagophores or isolation membranes. In yeast, autophagosome formation occurs in a specific place in close proximity to the ER and the vacuole named phagophore assembly sites (PAS) or pre-autophagosomal structures, where specific molecules are selectively recruited when autophagy is activated [132]. In mammals, phagophores grow in multiples places in close proximity to the ER, especially in the ring-like subdomains of this organelle, called omegasomes, which are enriched in phosphatidylinositol-3-phosphate (PI3P) and specific PI3P-binding proteins [133,134,135]. Direct continuities between the phagophore (and autophagosome) and the ER have been described [136], as well as tubular connections between the omegasome and the phagophore [137]. ER exit sites, ER–mitochondria contact sites, and ERGIC can also be the places of initiation. Different membranes have been proposed as sources for the nascent autophagosome, including the ER, mitochondria, Golgi complex, endocytic elements, and plasma membrane. The phagophore grows, curves around a portion of the cytoplasm, and finally closes, forming a double-membrane organelle called the autophagosome. Finally, it fuses with the lysosome and the inner membrane and the content is degraded, releasing simple molecules such as amino acids, lipids, and sugars [138]. Autophagosomes can also fuse with late endosomes, forming amphisomes, which also fuse with lysosomes [139].

Many aspects of the complex molecular machinery involved in this process are well known (for a detailed explanation, see [140,141]). This process is regulated by evolutionarily conserved autophagy-related (ATG) proteins. Twenty of these ATG proteins constitute the core machinery and can be grouped into different functional unities [142]. Initiation requires two kinase complexes: the unc-51 like kinase (ULK) complex (Atg1 complex in yeast) and the phosphatidylinositol 3-kinase class 3 (PI3KC3)/vacuolar protein sorting 34 (VPS34) complex. The ULK complex is the first inductor of the autophagic process. Upstream nutrient/energy-sensing kinases (e.g., the mammalian target of rapamycin complex 1, mTORC1) activate this complex, which is mostly cytosolic, and recruit to phosphatidylinositol synthase-enriched subdomains of the ER [143]. The activation of ULK phosphorylates downstream effectors, including ATG9, in vesicles [144] to induce autophagosome formation. ATG9 is the only membrane protein and, in mammals, is under steady-state conditions in the TGN/late endosomes; however, after autophagic activation, ATG9 vesicles move close to the autophagosome in their formation (the Atg9 compartment in yeast) [145]. ATG9 vesicles seem to provide membranes for the formation of the autophagosome, but also supply important regulators such as lipid-metabolizing enzymes. In fact, its role as a lipid scramblase has recently been demonstrated [146]. Soon after ULK, the PI3KC3/VPS34 lipid kinase complex is recruited, inducing the production of PI3P (at low levels in the ER in basal conditions), which recruits autophagy effectors that drive the formation of the omegasome: DFCP1 and WIPI proteins [133,147]. WIPI proteins recruit two ubiquitin-like conjugation systems that work in cascade: ATG7-ATG3 (acting as E1- and E2-like enzymes) and ATG12- 5/ATG16L1 (acting as E3-like enzyme) [148]. These systems enable the lipidation of microtubule-associated protein light chain 3 (LC3) isoforms, members of the Atg8 family of proteins. These ubiquitin-like proteins are usually unlipidated and cytosolic (LC3-I) but become covalently bound to the amino acid of phosphatidylethanolamine in the membrane of the phagophore (LC3-II). The last one is usually used as a marker of autophagic flux. The insertion of lipidated Atg8 proteins and ATG9-containing vesicles (which putatively supply lipids from different membranes) is responsible for the growth of the phagophore and its closure to form the autophagosome [149,150]. Atg8 proteins work as adaptors to recruit other factors that contribute to the membrane expansion of the autophagosomal membrane and the recruitment of selective autophagy receptors.

Autophagy operates as an intracellular membrane pathway. Thus, it is evident that it shares some regulatory mechanisms with other transport routes, including Rabs. Rab1/Ypt1 is an important regulator of autophagy, operating at early steps [151,152,153,154,155,156] (Figure 2B and Figure 3). In addition to Rab1, other Rabs are also known to be regulators of autophagy, including Rab33B (autophagosome elongation), Rab7 (fusion autophagosome-lysosome), Rab32 (autophagosome formation), Rab5 (autophagosome closure), Rab24 (co-localized with LC3), and Rab2 (autophagosome and autolysosome formation) [157,158].

In yeast, Ypt1 is activated by the Trs85 subunit of TRAPIII and regulates autophagy by recruiting the Atg1 complex to the PAS [159]. In mammals, Rab1 regulates the translocation of the ULK complex to the autophagosome formation sites [160]. In addition, it also regulates PI3P production at the omegasome. Thus, Rab1 is a specific activator of the VPS34 complex I (involved in autophagic sorting) but not II (involved in endocytic sorting) [161].

Ypt1 is activated by the TRAPPIII subunit Trs85 and also interacts with Atg11, which may be involved in the recruitment of Atg9 vesicles to PAS [154,162]. In mammals, Rab1 has been found in ATG9 vesicles and LC3-positive autophagosomes [152,163]. During starvation, ATG9 moves from Rab11-positive TGN/recycling endosomes to Rab1-positive ERGIC/Golgi membranes. This process is controlled by the interaction of TBC1D14 with the TRAPPC8 subunit (the mammalian ortholog of Trs85) of TRAPPIII [164]. There are several differences in the roles of Rab1 isoforms [165]. The depletion of Rab1A induces the mis-location of ATG9 and decreases omegasome formation. This route is independent of ER–Golgi transport. Thus, Rab1A specifically inhibits the very early steps of autophagy. In contrast, the depletion of Rab1B and Rab2 (another Rab involved in ER–Golgi transport) has the opposite effect of increasing autophagy. The significance of these differences remains to be established.

It has been proposed that there is a functional and spatial link between the secretory pathway and autophagy at the early stages, and the key component of this cross route may be Rab1. Under starvation conditions, there is a reduction in protein secretion and secretory compartments are redirected to the autophagy pathway. Autophagosomes are spatially and functionally linked to ERES [166]. Moreover, COPII vesicles and ERGIC can be sources of membranes for autophagy. Under stress conditions, ER-derived COPII vesicles, instead of being directed to the Golgi, can be a source of membranes for autophagosomes [85,167,168]. However, ERGIC can be the source of COPII vesicles after the activation of autophagy. ERGIC is a source of the small vesicles involved in LC3 lipidation, a process that requires the activation of phosphotidylinositol-3 kinase (PI3K) and the recruitment of COPII proteins in this compartment [169,170]. This process requires the remodeling of ERES, which is specifically mediated by some specific components of the UKL complex [171]. A critical step of this process is the phosphorylation of the COPII component Sec23B by ULK [172]. The ULK-dependent phosphorylation of COPII subunits also regulates ERES morphology and causes a reduction in secretion during active autophagy [173]. As an activator of ULK and an abundant component of the ERGIC, Rab1 may regulate this mechanism under stress conditions. TRAPPIII, the GEF for Rab1, binds the COP II coat subunit Sec23 at the PAS, which is necessary for autophagy [85]. In addition, Ypt1/Rab1 directly recruits the casein kinase 1 (CK1*δ*) family member Hrr25, a Golgi-localized serine–threonine kinase that regulates ER–Golgi traffic and autophagy, to COPII vesicles, activating its kinase activity and regulating the traffic of these vesicles in these two pathways [174,175].

## 6. Rab1, Secretory Pathway, and PD

Rab1A and Rab1B are expressed in SN, although at a lower level than other Rabs, such as Rab3A and Rab3C [71]. Interestingly, the levels of this Rab rose by 50% in human PD samples [46], as well as in a cellular model of PD [45]. Whether this overexpression is an attempt by the cell to rescue normality or is the result of cellular damage is not known.

The first connection between Rab1 and PD was observed in a yeast model [176]. Researchers have found that the main toxic effect of wild-type and PD-related mutant α-syn is due to the alteration of ER–Golgi transport and the best suppressor of this toxicity was found to be Ypt1. A few other proteins inhibit this toxicity, and, interestingly, all of them are involved in the same transport step, including Gyp8, which is the GAP for Ypt1. This effect is not restricted to this simple cellular model; is has also been shown that Rab1 protects against the mammalian dopaminergic neurons’ death induced by *α*-syn. For the first time, intracellular trafficking, and, more specifically, ER–Golgi trafficking, has been related to the pathogenesis of PD. In yeast, only this Rab rescues *α*-syn. However, Rabs associated with later transport steps in neuronal models of PD (i.e., Rab3A and Rab8) have been shown to protect against α-syn toxicity [177]. In mammalian cells, the overexpression of wild-type or A53T mutant *α*-syn inhibits COPII vesicles’ docking and fusion with early Golgi by the inhibition of the SNARE complex assembly [178]. The pathological accumulation of α-syn affects hydrolase trafficking and therefore lysosomal activity, and this dysfunction can be improved by Rab1 overexpression [179]. This alteration is due to the aberrant interaction of α-syn with GM130, which disrupts Rab1-dependent ER–Golgi traffic and the Golgi structure.

## 7. Relationship between PD-Associated Golgi Fragmentation and Rab1

GC is a cellular organelle that acts as the central station of intracellular transport. It is involved in the post-translational modification (mainly adding sugar moieties) of proteins and lipids synthesized in the ER. A network of tubule–vesicular elements associated with the trans side, the TGN, is involved in the classification, packaging, and delivery of these products to the final destination in or out of the cell. It is especially developed in cells with synthetic activity, as is the case of cells of neurons. It is commonly found in a perinuclear position, showing a half-moon disposal. Under the electron microscope, it appears as stacked cisternae, classified according to their position in the organelle as cis, medial, and trans cisternae. In mammalian cells, Golgi stacks are laterally linked, forming a so-called Golgi ribbon [180,181,182].

Interestingly, the Golgi complex of neurons appears to be fragmented in most neurodegenerative diseases, including AD and PD [45,46,183]. The Golgi ribbon is broken and stacks appear dispersed throughout the cytoplasm. Even though it has been demonstrated that the GC remains functional after fragmentation, the ribbon structure is implicated in high-level functions, such as mitosis entry, apoptosis regulation, stress response, axodendritic polarity, and cell migration [184,185]. Therefore, Golgi fragmentation significantly alters neural physiology, inducing traffic failure and thereby causing a deficiency in the dopaminergic transport in PD, resulting in the limited capture of this neurotransmitter in vesicles and its limited accumulation in the cytosol. Toxic dopamine could aggregate and modify α-syn and increase its toxicity, which could lead to the formation of LBs and, finally, cell death [186].

The maintenance of the GA cytoarchitecture depends on many structural (including golgins and Golgi reassembling stacking proteins, GRASPs) and regulating proteins (Rabs, SNARE, etc.) and the cytoskeleton. The alteration of these elements may be the cause of this fragmentation. In addition, the alteration in intracellular transport caused by AG fragmentation has been shown to cause deficits in cell degradation activity due to the deficiency of lysosomes with proteases such as cathepsin D, which facilitates the degradation of α-syn [187]. Previous studies support the notion that Golgi fragmentation occurs before the formation of α-syn aggregates [45]. In fact, it is the alteration of Rab1-dependent ER-to-Golgi transport that causes imbalances in the traffic into and out of the GC. Rab1 is also involved in intra-Golgi transport, although it has not been detected in medial/trans cisternae. It is important for the Golgi structure, because the downregulation or microinjection of mutants induces Golgi alterations [188,189,190]. The increased amount of Rab1a observed in survival nigral cells is theorized to be an attempt to restore the Golgi’s structure during fragmentation. Thus, Rab1 rescues Golgi fragmentation in dopamine cells overexpressing α-syn and, interestingly, also improves the control of motor function in hemiparkinsonian rats [183]. The alteration of Rab1-dependent ER–Golgi transport seems to be a common cause of Golgi fragmentation in several neurodegenerative diseases [185].

## 8. Rab1, Autophagy, and PD

Altered autophagy seems to be a common feature of neurological diseases. The deletion of genes coding for essential proteins in autophagosome formation, such as Atg5 [191] or Atg7 [192], causes neurodegeneration. In AD brains, autophagosomes and cathepsin-containing autophagolysosomes are common in neurite processes, including synaptic ends [193]. Autophagic vacuoles containing cytoplasmic materials are also common in nigral neurons affected by PD [194]. Interestingly, Lewy bodies are not observed in cells with high amounts of these vacuoles. Lewy bodies and neurites contain autophagic markers, and the lipidated form of LC3 (LC3-II) is highly expressed in PD nigral samples [195,196]. Impaired autophagy may contribute to the development of sporadic PD. Thus, a deficit of ATG7 in dopamine neurons induces high alterations in these cells but delays neurodegeneration and the appearance of late-onset locomotor deficits in mice models [197]. The whole-brain loss of this ATG protein induces the presynaptic accumulation of α-syn and LRRK2.

LRRK2 and α-syn have been extensively studied in relation to their involvement with aberrant autophagy. Mutations in these PD-related genes affect autophagic function [198]. Wild-type α-syn contains a KFERQ sequence motif, so it is degraded by chaperone-mediated autophagy. Conversely, autophagy is the main mechanism for the degradation of fibrillary *α*-syn, while the proteasome system is preferred for phosphorylated *α*-syn oligomers [199]. Some PD-related mutations affect this pathway. Thus, A53T and A30P mutations block chaperone-mediated autophagy but enhance autophagy, which may be a compensatory response [200]. Moreover, autophagy is increased in samples of PD patients (with or without A53T mutations) [201]. In contrast, another study supports the notion that the overexpression of the A53T mutant inhibits autophagy, resulting in the accumulation of this defective protein [202]. In addition, *α*-syn can affect autophagosome formation by altering the actin cytoskeleton. It induces an excess of stabilization, which is dependent on Arp2/3, which was found to impair autophagosome maturation and mitophagy in a Drosophila model of PD [203].

The wild-type form of LRRK2 is degraded by chaperone-mediated autophagy or the ubiquitin–proteasome system but not autophagy [204]. LRRK2 is a negative regulator of autophagic activity and the expression of PD-related mutations and induces abnormal autophagy [205]. The G2019S mutation, the most common PD-related mutation that increases kinase activity, induces autophagy abnormalities and affects neurite development [206,207]. This mutation induces defective autophagy by affecting autophagosome transport and maturation [208]. Many Rabs are substrates of LRRK2 (Rab3A/B/C, Rab3D, Rab8A, Rab8B, Rab10, Rab12, Rab29/RAB7L1, Rab35, and Rab43) [209], and all PD-related mutations enhance Rab phosphorylation [210]. Interestingly, it was recently demonstrated that Rab1 is also an endogenous substrate of this kinase [211] and might be the mechanism used for autophagy regulation.

A relationship between Rab1, autophagy, and neurodegenerative diseases has been described [160]. A GGGGCC hexanucleotide repeat expansion in the C9orf72 gene is the most common genetic cause of amyotrophic lateral sclerosis and frontotemporal dementia (C9ALS/FTD) and is also associated with sporadic forms. C9orf72 is a Rab1 effector that regulates the translocation of the ULK complex in the initial steps of autophagy. This link is also evident in PD. The overexpression of *α*-syn inhibits autophagy by inhibiting Rab1A, which results in the mislocalization of Atg9 and a reduced number of omegasomes [165]. Moreover, the overexpression of Rab1A rescues the effects of the overexpression of *α*-syn.

## 9. Concluding Remarks

Rab1 controls the early steps of the secretory pathway and autophagy, as summarized in Figure 1. The overexpression of this GTPase by survival nigral neurons can have the effect of minimizing PD abnormalities. An elevation of the secretory flux may help to restore the reduction in dopamine traffic to the terminal end and the lysosomal defects caused by the alteration of *α*-syn-dependent ER to Golgi trafficking [176]. In addition, it also may restore the Golgi ribbon architecture, which affects polarized transport in neurons [185]. In fact, Rab1 overexpression was found to ameliorate motor symptoms in a PD model [183]. The activation of autophagy has been proposed as a treatment for PD. However, the results found are contradictory. The autophagy-stimulating antibiotic rapamycin prevents the loss of dopamine neurons [212]. Impaired autophagy induces progressive dopamine cell loss but, conversely, enhances dopamine neurotransmission and ameliorates motor symptoms [21]. It is possible that unbalanced autophagy could result in cell stress and, finally, death [213]. The overexpression of Rab1 (at least Rab1A) enhances autophagy and restores the defects produced by *α*-syn [165]. In fact, our group found that it reduced the number of *α*-syn aggregates in a cellular model of PD (Figure 4). Recent research has focused on *α*-syn as a therapeutic target for PD in an attempt to reduce the levels of production and aggregation or increase the degradation [214]. Rab1, as a regulator of *α*-syn toxicity and substrate of LRRK2, may also be a good candidate. Rab1 expression/function (and their binding partners) may be altered using small molecules that interact with selective regions of this protein that physically interact with regulators/effectors [215]. The search for therapeutic agents modulating Rab1-dependent *α*-syn toxicity is currently underway [216]. Alternatively, the function of Rab1 may be modified by targeting the regulatory proteins, including GEF, GAPs, and GDIs. Several compounds have been tested in attempts to inhibit the disease-dependent activation of members of the Ras superfamily of small GTPases (including Rabs), although these investigations are only at the initial laboratory/preclinical stage [217]. This inhibition can be achieved by the inhibition of GEF activity, GEF–GTPase interaction, or GAP activity or through the stabilization of the GTPase–GDI complex, among other strategies. However, it should be taken into consideration that most of these regulatory proteins are not specific for a single GTPase. Given that most data support the notion that Rab1 activation has a positive effect on dopamine neurons, it is necessary to look for specific Rab-activating compounds. As indicated above, the prenylation of Rab by isoprenoid groups is necessary for membrane location and interaction with effectors, meaning that this molecular mechanism can be also used as a regulator of Rab function. Isoprenoids are produced by the mevalonate pathway, which is also involved in cholesterol biosynthesis. Some inhibitors of this route have been successfully used as therapeutic approaches for type 2 diabetes, given that Rab GTPases (including Rab1) are necessary for glucose homeostasis [218]. Statins, which are well-known blockers of cholesterol generation, are inhibitors of hydroxymethyl-3-methylglutaryl coenzyme A, a critical enzyme of this pathway. The effects of statins in PD patients have been tested, although the results gained are conflicting [219]. Again, this approach does not seem to be as specific as needed. New experiments in cellular and animal models will be necessary to clarify the role of Rab1 in the development of PD and elucidate how specific compounds affect *α*-syn aggregation, Golgi structure, secretory traffic, and autophagy in dopamine neurons. This small GTP is a good example of how basic research may help us to understand complex pathologies and search for new therapeutic approaches.

## Figures and Tables

**Figure 1 ijms-22-12087-f001:**
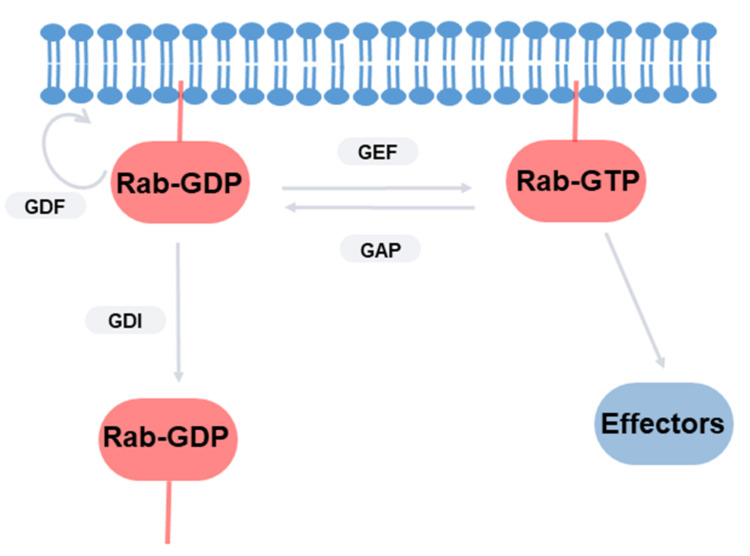
Regulation of GTP–GTP cycle of Rab GTPases by guanine nucleotide exchange factors (GEFs), GTPase-activating proteins (GAPs), GDP dissociation inhibitor (GDI), and GDI displacement factors (GDF). Membrane-associated GTP-bound Rabs are able to interact with effectors.

**Figure 2 ijms-22-12087-f002:**
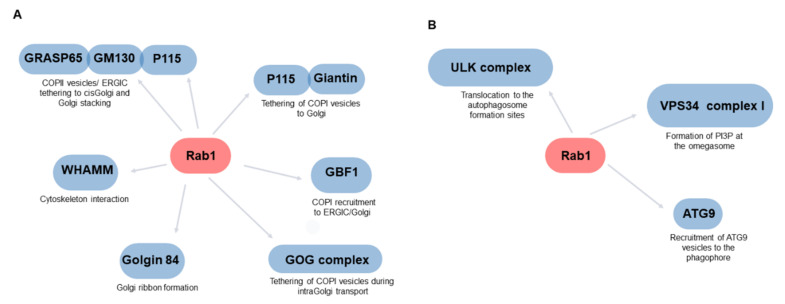
Main effectors of Rab1 involved in secretory traffic (**A**) and autophagy (**B**).

**Figure 3 ijms-22-12087-f003:**
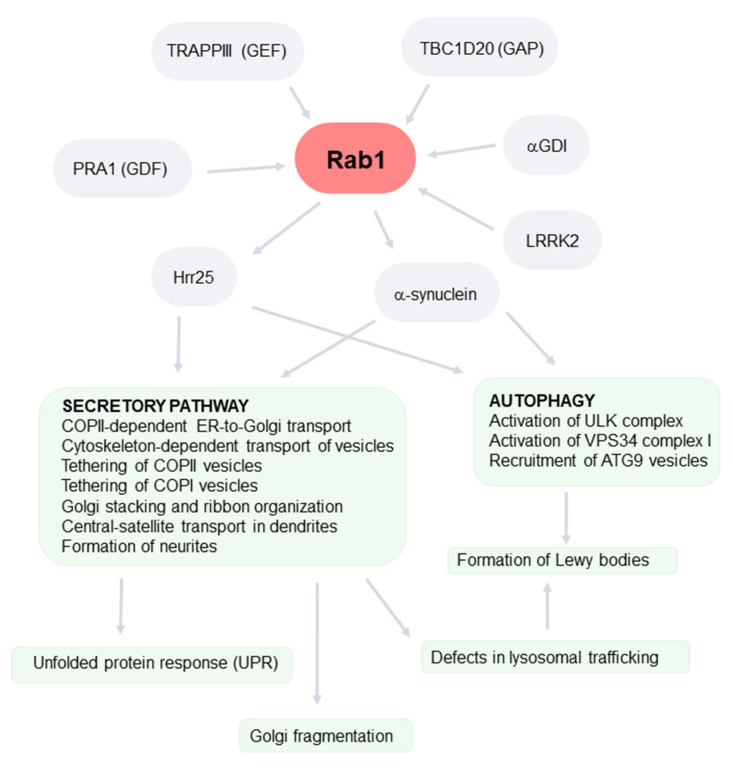
Overview of the roles and regulation of Rab1 and how it may be affected in PD. Activity and association with membranes of Rab1 are regulated by TRAPPIII (a guanine nucleotide exchange factor, GEF), TBC1D120 (a GTPase-activating protein, GAP), PRA1 (a GDI displacement factor, GDF), and αGDI (a GDP dissociation inhibitor, GDI). This activity is also regulated by the PD-related protein LRRK2. Rab1 regulates the early steps of the secretory pathway and autophagy through many effectors (see the full text for details). In addition, it regulates Hrr25, a member of the casein kinase 1 family, which also regulates these pathways. The expression of mutant forms of α-synuclein or overexpression of the wild type impair the secretory pathway, resulting in the activation of the UPR, the fragmentation of the Golgi ribbon, and the failure of the lysosomal route. These PD-associated alterations also affect autophagy. PD-related mutants of LRRK2 may enhance the activity of this GTPase, impairing secretory and autophagy pathways. Unbalanced secretory/autophagic/endolysosomal routes may lead to the formation of Lewy bodies.

**Figure 4 ijms-22-12087-f004:**
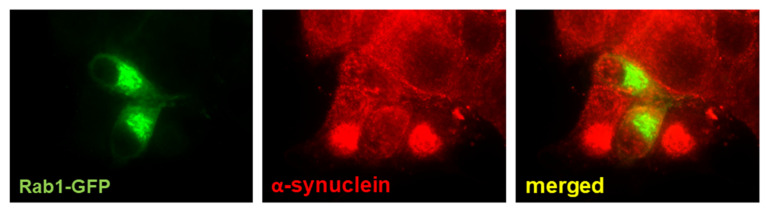
Rab1 overexpression reduces *α*-synuclein aggregates. These aggregates were induced in neuron-like differentiated PC12 cells by treatment with METH as a model of PD. Cells overexpressing Rab1-GFP have a reduced number of aggregates.

## Data Availability

Not applicable.

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
