# Peer review of "Focus on the Small GTPase Rab1: A Key Player in the Pathogenesis of Parkinson’s Disease"

_ijms, 2021, doi:10.3390/ijms222112087_

Round 1

Reviewer 1 Report

Dear Authors

the paper is well organized and presented.

Author Response

Thank you very much for your positive comments

Reviewer 2 Report

The article “Focus in the small GTPase Rab1: a key player on the pathogenesis of_ Parkinson’s disease” provides insight into the role of Rab1 in PD. This is a very useful and comprehensive review with nicely covered topic. This review overviews our current understanding of the topic, however I would like to see more potential knowledge gaps in the field that might require immediate investigations for the discovery of novel drug targets for the prevention and treatment of neurodegenerative diseases, including PD. 

I also suggest to add figures and tables that could be particularly helpful – they would make it possible to manage a multiple, very complicated relationships.

I also recommend to add information how posttranslaional modifications (e.g., prenylation) may affect pathogenesis of PD and commentary if targeting this step could be useful (like in the case of diabetes (e.g., doi: 10.1021/acs.jmedchem.1c00410).

Minor points about the article are below:

This small protein (14 Kda): shoud be kDa

Symbols for genes are italicized, whereas symbols for proteins are not italicized (authors should correct this issue, e.g., the SNCA gene

Author Response

Reviewer 2

Point 1. The article “Focus in the small GTPase Rab1: a key player on the pathogenesis of_ Parkinson’s disease” provides insight into the role of Rab1 in PD. This is a very useful and comprehensive review with nicely covered topic. This review overviews our current understanding of the topic, however I would like to see more potential knowledge gaps in the field that might require immediate investigations for the discovery of novel drug targets for the prevention and treatment of neurodegenerative diseases, including PD. 

Response: Thank you very much for your comments. The manuscript has been improved with new information on potential research lines (see below)

Point 2. I also suggest to add figures and tables that could be particularly helpful – they would make it possible to manage a multiple, very complicated relationships.

Response: New figures have been added. New figure 1 explains the regulation of GTP-GDP cycle of Rabs. New figure 2A and B describes Rab1 effectors and the relationship with secretory pathway and autophagy, respectively.

Point 3. I also recommend to add information how posttranslaional modifications (e.g., prenylation) may affect pathogenesis of PD and commentary if targeting this step could be useful (like in the case of diabetes (e.g., doi: 10.1021/acs.jmedchem.1c00410).

Response: Thank you very much for this suggestion. This point has been developed in the new version of the manuscript (lines 547-569). Regulation of Rab prenylation and the relationship with PD have been described. In addition, we added new information about compounds affecting Rabs regulatory proteins, including GEF, GAPs, and GDIs.

4.- Minor points about the article are below:

This small protein (14 Kda): shoud be kDa

Symbols for genes are italicized, whereas symbols for proteins are not italicized (authors should correct this issue, e.g., the SNCA gene

Response: Done

Reviewer 3 Report

The review describes the role of GTPase Rab1 in Parkinson’s Disease. In particular, the authors summarized in depth the key events of GTPase Rab1 function in dopamine neurons and its implications in dynamics of the neurodegenerative processes of Parkinson’s Disease.

The review is interesting and updated with recent references of experimental evidences. However, the authors could improve some aspects of this review:

They should shorten section 1 Brief introduction to PD pathogenesis. Some information is redundant. The reviewer suggests to delete “Brief” from the title of this section.

The Figure 1 could be considered a summary of the interactions of Rab1 with critical targets for the Parkinson’s Disease. In this regard, the authors should help the reader with new tables and figures for some specific aspects of the review which help to interpret the information in Figure 1. Example: a figure with general molecular mechanisms of Rabs (section 2. Rab GTPases); a table with the different Rabs and their localization and function at the neuronal level (section 2. Rab GTPases); a figure with the relevant molecular interaction of Rab1 at ER levels (section 4. The role of Rab1 in the secretory pathway) or other cellular structures involved in the secretory pathway, similarly also for autophagy pathway.

Author Response

Reviewer 3

Point 1. The review describes the role of GTPase Rab1 in Parkinson’s Disease. In particular, the authors summarized in depth the key events of GTPase Rab1 function in dopamine neurons and its implications in dynamics of the neurodegenerative processes of Parkinson’s Disease.

The review is interesting and updated with recent references of experimental evidences. However, the authors could improve some aspects of this review:

Response: Thank you very much for your comments and suggestions

Point 2. They should shorten section 1 Brief introduction to PD pathogenesis. Some information is redundant. The reviewer suggests to delete “Brief” from the title of this section.

Response: “Brief” has been removed from the title. Some sentences of the introduction section have been removed in order to shorten this section.  We believe that the information in this section is completely necessary for readers with are not expert in PD. 

Point 3. The Figure 1 could be considered a summary of the interactions of Rab1 with critical targets for the Parkinson’s Disease. In this regard, the authors should help the reader with new tables and figures for some specific aspects of the review which help to interpret the information in Figure 1. Example: a figure with general molecular mechanisms of Rabs (section 2. Rab GTPases); a table with the different Rabs and their localization and function at the neuronal level (section 2. Rab GTPases); a figure with the relevant molecular interaction of Rab1 at ER levels (section 4. The role of Rab1 in the secretory pathway) or other cellular structures involved in the secretory pathway, similarly also for autophagy pathway.

Response: New figures have been added as suggested by the reviewer. New figure 1 explains the regulation of GTP-GDP cycle of Rabs. New figure 2A and B describes Rab1 effectors and the relationship with specific steps of the secretory pathway and autophagy. However, we do think that a table with all neuron-associated Rabs is outside of the scope of this review. First, we focus only in one specific Rab and a detailed information about other Rabs seems to be unnecessary. Second, this information can be found in recent reviews cited in text (see references 58-62). Finally, this table will give too much information for this review and, perhaps, deserves a special review. In fact, we try to elaborate this table and our preliminary version employ many pages and dozens of new original references. Please note that there are at least 40 Rabs with well-known specific functions in neurons.

Round 2

Reviewer 2 Report

The authors improved the manuscript as requested.